# Deep Learning-Based Energy Expenditure Estimation in Assisted and Non-Assisted Gait Using Inertial, EMG, and Heart Rate Wearable Sensors

**DOI:** 10.3390/s22207913

**Published:** 2022-10-18

**Authors:** João M. Lopes, Joana Figueiredo, Pedro Fonseca, João J. Cerqueira, João P. Vilas-Boas, Cristina P. Santos

**Affiliations:** 1Center for MicroElectroMechanical Systems (CMEMS), University of Minho, 4800-058 Guimarães, Portugal; 2LABBELS—Associate Laboratory, 4710-057 Braga/4800-058 Guimarães, Portugal; 3Porto Biomechanics Laboratory (LABIOMEP), Faculty of Sports, University of Porto, 4200-450 Porto, Portugal; 4Life and Health Sciences Research Institute (ICVS), School of Medicine, University of Minho, 4710-057 Braga, Portugal; 5Faculty of Sports and CIFI2D, University of Porto, 4200-450 Porto, Portugal

**Keywords:** artificial intelligence, deep learning, energy expenditure, gait rehabilitation, human-in-the-loop, robotics-based rehabilitation, wearable sensors

## Abstract

Energy expenditure is a key rehabilitation outcome and is starting to be used in robotics-based rehabilitation through human-in-the-loop control to tailor robot assistance towards reducing patients’ energy effort. However, it is usually assessed by indirect calorimetry which entails a certain degree of invasiveness and provides delayed data, which is not suitable for controlling robotic devices. This work proposes a deep learning-based tool for steady-state energy expenditure estimation based on more ergonomic sensors than indirect calorimetry. The study innovates by estimating the energy expenditure in assisted and non-assisted conditions and in slow gait speeds similarly to impaired subjects. This work explores and benchmarks the long short-term memory (LSTM) and convolutional neural network (CNN) as deep learning regressors. As inputs, we fused inertial data, electromyography, and heart rate signals measured by on-body sensors from eight healthy volunteers walking with and without assistance from an ankle-foot exoskeleton at 0.22, 0.33, and 0.44 m/s. LSTM and CNN were compared against indirect calorimetry using a leave-one-subject-out cross-validation technique. Results showed the suitability of this tool, especially CNN, that demonstrated root-mean-squared errors of 0.36 W/kg and high correlation (ρ > 0.85) between target and estimation (${\bar{R}}^{2}$ = 0.79). CNN was able to discriminate the energy expenditure between assisted and non-assisted gait, basal, and walking energy expenditure, throughout three slow gait speeds. CNN regressor driven by kinematic and physiological data was shown to be a more ergonomic technique for estimating the energy expenditure, contributing to the clinical assessment in slow and robotic-assisted gait and future research concerning human-in-the-loop control.

## 1. Introduction

Gait disabilities are among the most frequent disabilities in European countries [1]. Either caused by aging or cardiovascular and/or neurological disorders, impaired gait strongly affects the walking energetic efficiency of persons [2]. Therefore, energy expenditure has gained importance in gait rehabilitation, being a golden marker of gait quality and a primary outcome for evaluating manual or robotics-assisted therapies. It is usually assessed by evaluating the exchanges of oxygen consumption ($\dot{V}O_{2}$) and carbon dioxide production ($\dot{V}{CO}_{2}$) through indirect calorimetry [3], using wearable gas analyzers. Then, these exchanges of gas are commonly translated into energy using the Brockway’s equation [4].

Powered assistive devices, namely exoskeletons and orthoses, guide users for better and faster functional recovery by introducing assist-as-needed control strategies [5,6,7,8,9]. Energy expenditure has been used in several studies to assess the effectiveness of using such assistive devices [10,11,12,13,14]. For instance, Awad et al. [12] demonstrated that using powered assistive devices in the gait rehabilitation of post-stroke survivors improves their energy efficiency by 10% regarding unpowered assistance, which the authors found to be equivalent to a 32% reduction in the metabolic cost associated with post-stroke walking.

Recent trends point toward the use of exoskeletons and orthoses to reduce the energetic effort and improve the ability of impaired subjects to independently perform activities of daily living through the human-in-the-loop control [15]. This assist-as-needed control strategy evokes personalized and user-oriented assistance [16] by using the energy expenditure as a cost function for optimizing the robotic assistance in real-time [16,17,18]. Thus, the human-in-the-loop control requires accurate and timely energy expenditure estimates to effectively tailor the robotic assistance to reduce subject’s energy expenditure. However, the use of indirect calorimetry may be a problem for control purposes, given its noisy and delayed data [17], hindering the optimization process. Moreover, it is not the most ergonomic setup to use in persons with disabilities since it entails a minimum of invasiveness that should be avoided and may increase their energetic effort.

To overcome these disadvantages, new approaches involving artificial intelligence (AI) algorithms are being studied to attain generalized regression models that timely estimate energy expenditure using easy-to-obtain inputs. These inputs are obtained using smaller and more ergonomic sensors that feed data to regressors, such as linear regression models and neural networks, to replace the use of gas analyzers, which gives noisy and delayed data and can be uncomfortable in long term use. For instance, Ingraham et al. [19] presented a linear regression model to estimate energy expenditure of level, incline, and backward walking, running, cycling, and stair climbing. The authors used multimodal data tracked by wearable sensors, namely breath rate (BR), minute ventilation (MV), heart rate (HR), oxygen saturation (SpO_2_), acceleration, electrodermal activity, skin temperature, and electromyography (EMG). They have shown reasonable performance, presenting root-mean-squared errors around 1.0 W/kg for the best model. Beltrame et al. [20] used HR, BR, MV, hip acceleration, walking cadence, and HR variation (∆HR) as estimators for a random forest regression model. The results revealed a high correlation with the ground truth signal (*r* ≥ 0.69). Beltrame et al. [21] also explored the feasibility of a single multilayer perceptron (MLP) to estimate oxygen consumption using the HR, gait speed and grade, rest and activity time, gender, and body mass. The authors found a high correlation between the estimated and target signal (*r* > 0.90). Zhu et al. [22] explored the potential of a convolutional neural network (CNN) to estimate the energy expenditure when compared to MLP and an activity-specific linear regressor, and the authors found an improvement of more than 30% when using deep learning regressors. 

However, the study of AI algorithms’ effectiveness for estimating the energy expenditure at slow gait speeds remains underexplored. These slow gait speeds, often observed in persons with gait disabilities (gait speed ranges from 0.15 to 1.0 m/s, with an average of 0.46 m/s [23]), can entail lower variance in energy expenditure, hindering the estimation process. Moreover, there is still limited evidence of AI algorithms’ feasibility to estimate energy expenditure during robot-assisted gait. Slade et al. [15] estimated the energy expenditure of healthy users for ankle assisted walking at normal speed (1.25 m/s), using ground reaction force (GRF) and EMG signals as inputs. However, GRF signals require force platforms, which are non-wearable nor a setup often available in rehabilitation centers or hospitals. To the authors’ best knowledge, there is no available regression algorithm that relies only on data from ergonomic on-body sensors to accurately estimate energy expenditure while walking in robotic-assisted and slow gait conditions. 

To attain these challenges, this work presents a deep learning regression tool for steady-state energy expenditure estimation using estimators tracked by ergonomic, reliable, and clinical-accepted on-body sensors. Novelty arises from a tool that can estimate energy expenditure in both assisted and non-assisted walking by an ankle exoskeleton while considering slow gait speeds (0.22, 0.33, and 0.44 m/s). For this purpose, we compared two deep learning regression networks, namely the long short-term memory (LSTM) and CNN, attending to the results of Zhu et al. [22], using HR, lower limb kinematic, and EMG data as estimators. This study hypothesizes that a deep learning regressor fed by fused gait-related biomechanical and physiological data enables an accurate and timely energy expenditure assessment in basal, assisted, and non-assisted walking, even for slow speeds. We studied this hypothesis through the leave-one-subject-out cross-validation (LOOCV) technique by comparing the estimation of deep learning tool against indirect calorimetry and verify its feasibility. This study presents two-fold contributions: (i) in terms of clinical assessment, it proposes a more ergonomic and rapid technique to estimate the energy expenditure, which is reliable in slow gait, as commonly observed in gait impaired subjects; (ii) in terms of rehabilitation robotics, it supports future research insights regarding human-in-the-loop control by accurately and efficiently estimating the energy expenditure in robotic-assisted conditions through timely and reliable acquired data.

## 2. Materials and Methods

The development of the deep learning-based tool required a prior data collection. This was conducted under the ethical procedures of the Ethics Committee in Life and Health Sciences (CEICVS 006/2020), following the Helsinki Declaration and the Oviedo Convention. All participants gave their informed consent to be part of the study. Data were collected at the LABIOMEP, Porto Biomechanics Laboratory, University of Porto.

### 2.1. Participants

Eight healthy participants (four females and four males, see Table 1) were recruited and accepted to participate in this work. A list of eligibility criteria was outlined to conduct the experimental data collection. All subjects that had: (i) 18 or more years, (ii) body mass within 45 and 90 kg, and (iii) height within 150 and 190 cm were included in the study. The inclusion criteria regarding the anthropometric data were imposed given the exoskeleton’s inherent requirements. The subjects were excluded if they reported any disturbance of locomotion or balance, caused by any known neurological or musculoskeletal injury. Table 1 presents the participants’ detailed anthropometric data.

### 2.2. Instrumentation

The participants were instructed to wear shorts and standard sports shoes for better sensor accommodation. Subsequently, we instrumented each participant with the following sensor setup. First, the participants were instrumented with a wireless Polar H10 (Polar Electro Oy, Kempele, Finland, validated in ref. [24]) at the chest, that was used to monitor the HR. Second, they were instrumented with eight wireless EMG surface electrodes from the Trigno^TM^ Avanti Platform (Delsys, Natick, MA, USA) on the *tibialis anterior* (TA), *gastrocnemius lateralis* (GL), *bicep femoris* (BF), and *vastus lateralis* (VL) muscles of both legs. The sensors were placed following the Surface ElectroMyoGraphy for the Non-Invasive Assessment of Muscles (SENIAM) recommendations [25], and fixed with a white strap, as illustrated in Figure 1. Third, we instrumented the participants with the wireless motion tracker system MVN Awinda (Xsens Technologies B.V., Enschede, The Netherlands, validated in ref. [26]), placing inertial measurement units (IMUs) on the feet, lower-leg, upper-leg, pelvis, and torso. Each IMU was secured with a black strap, as illustrated in Figure 1. Fourth, each participant was instrumented with a K4b^2^ metabolic respiratory sensor (COSMED, Rome, Italy, validated in ref. [27]), covering the facial respiratory ways, to measure $\dot{V}O_{2}$ and $\dot{V}{CO}_{2}$. Lastly, the participants were instrumented with an electrical ankle-foot exoskeleton from the H2-Exoskeleton (Technaid S.L., Madrid, Spain) in the lateral side of the right lower limb. This device provides one degree-of-freedom in the sagittal plane.

### 2.3. Experimental Protocol

Immediately after the placement of the EMG sensors, the participants performed three maximum voluntary contractions (MVC) for each muscle to normalize EMG data. For the BF muscle, the participants laid on a stretcher in ventral decubitus position with their knee slightly bent. One researcher immobilized the participants’ shank and asked them to perform maximum knee flexion. Regarding the VL muscle, the participants sat down on the same stretcher performing 90 degrees between thighs and shanks. The researcher immobilized their shank and asked the participants to perform maximum knee extension. For the TA and GL muscles, the participants laid on the stretcher, assuming dorsal decubitus position. To perform the MVC, the researcher immobilized the participants’ foot and asked them to perform maximum dorsiflexion of the ankle articulation, in the case of TA muscle, and maximum plantar flexion, in the case of GL muscles. This procedure was repeated for both legs. Afterward, the MVN Awinda sensors were placed and the MVN biomechanical model was calibrated by following the software guidelines: each participant assumed the *N*-pose, which refers to a neutral position of body segments, in upright position, looking forward with the two arms stretched and hands near the thighs. Each participant held this position for four seconds, and then walked forward in a normal fashion, turned, and walk backwards to the initial position. Subsequently, some anthropometric data, namely gender, height, and body mass, were introduced in the respirometer. After this, each participant experienced a one-day protocol consisting of two sessions: one session with and another without the orthotic device. In both sessions, the subjects performed three walking trials of twelve minutes, one per each gait speed: 0.22, 0.33, and 0.44 m/s. In each trial, the participants were instructed to remain in standing position during the first three minutes to assess the basal energy expenditure, followed by a walking activity on a treadmill that lasted six minutes, and finishing with three minutes in standing position as a recovering period. The participants rested for ten minutes between each trial and/or session. In the assisted session, the ankle-foot exoskeleton assisted according to a position control strategy.

### 2.4. Data Acquisition

Data acquisition included: (i) $\dot{V}O_{2}$ and $\dot{V}{CO}_{2}$, in mL/sec, with the K4b^2^ respiratory sensor breath-by-breath; (ii) the muscular activation of the TA, GL, BF, and VL muscles for both lower limb at 1000 Hz using the 8-channel Trigno^TM^ EMG sensors and the Delsys acquisition software; (iii) the acceleration and angular velocity of lower-leg, upper-leg, feet, pelvis, and torso at 100 Hz using the MVN Awinda; and (iv) the heart rate using a Polar H10. Data acquisition commenced simultaneously for all devices, ensuring time synchronization.

### 2.5. Data Processing

The energy expenditure (*EE*), in Watts, was calculated using the $\dot{V}O_{2}$ and $\dot{V}{CO}_{2}$ following Brockway’s equation [4], depicted in (1).
(1)$$EE=16.58\;\dot{V}O_{2}+4.51\;\dot{V}{CO}_{2}$$


We average each activity (i.e., standing for three minutes, walking for six minutes, and recovering for three minutes) to assess the steady-state energy expenditure considering the last three minutes of each [28]. Due to the high noise of indirect calorimetry, a 95% confidence interval was calculated to eliminate possible outliers. Additionally, the steady-state energy expenditure was normalized by body-mass, as performed in [10,11,12]. Following these processing techniques, a step-like signal of energy expenditure was obtained (Figure 2A–D).

The HR signal was processed by following the same approach. Additionally, the HR was normalized considering the maximum HR expected for the individual considering his/her age, in years.

The kinematic data, namely the acceleration and angular velocity, were first filtered with a fourth-order zero-lag Butterworth filter with a cut-off frequency of 5 Hz [29]. Afterward, the total acceleration and angular velocity (vector’s magnitude) were calculated and low-pass filtered at 0.1 Hz to preserve the low frequencies that belong to the start/stop walking transitions. 

EMG signals were processed as follows. First, they were filtered with a zero-lag band-pass filter with cut-off frequencies of 20 and 450 Hz [30,31]. Second, we calculated the envelope using a low-pass filter with a cut-off frequency of 2 Hz and normalized considering the user’s MVC. Third, the signals were low-pass filtered at 0.1 Hz to preserve the low frequencies related to the start/stop walking transitions. Additionally, we calculated the EMG sum (Figure 2B), similar to Ingraham et al. [19], for both lower and upper leg muscles, to have a more general information regarding the muscles’ activation. Figure 2 illustrates an example of the post-processed data that were used to estimate the steady-state energy expenditure. 

### 2.6. AI-Based Regression Models

As regression models, we explored LSTM, suited for sequential data [32], and CNN, due to the reliable performance reported in Zhu et al. [22]. For both models, we fused the following estimators: (i) kinematic data, namely the acceleration (Figure 2A) and angular velocity (Figure 2B) of lower leg, upper leg, feet, pelvis, and torso; (ii) lower and upper leg EMG sum (Figure 2C); (iii) HR (Figure 2D); (iv) gait speed; and (v) anthropometric data, as in [21], as the gender (corresponding to a binary signal), age, and height. The input signals were rescaled between [−1, 1] considering a min-max algorithm.

Regarding the LSTM, we implemented the following architecture. For the first layer, we set the *sequence input layer* with a sequence length of 300 samples. In the second layer, we implemented one *LSTM layer* and studied the best number of neurons, considering 10, 50, 100, 150, and 200 neurons. To the best model found in the last step, we studied the effect of adding a second *LSTM layer* and ranged the number of neurons from 10 to 100 neurons. We introduced one *dropout layer* (*p* = 0.5) after each LSTM layer. As the penultimate layer, we added one *fully connected layer* to the best model found. The last layer is the *regression output layer*. Figure 3 illustrates an example of a LSTM neural network that was explored in this work.

For the CNN, we firstly implemented the *input layer*. At the second layer, we added one *convolution layer* with 8 filters, and we ranged the filter’s size from 5 [22] to 15 with a resolution of 5. We also tested increasing the number of filters to 16 and introducing a second *convolution layer*. The filters were initiated with the Glorot initializer and then optimized during the learning process. We used the *ReLu layer* as the activation function and an *average pooling layer* of size 2 for each *convolution layer*. We also introduced one *dropout layer* (*p* = 0.5) and one *fully connected layer*, ranging the number of neurons until a decrease in performance was detected. The last layer is the *regression output layer*. Figure 4 illustrates one example of CNN configuration that was studied in this work.

For both CNN and LSTM, we used the dropout layer and the L2 regularization method (λ = 0.0001) to prevent the existence of overfitting and gradient vanishing. These regression models were trained with the Adam optimization algorithm using mean-square error (MSE) as the loss function, considering the initial learning rate set to 0.01. We recursively implemented several models by changing the neural networks’ hyperparameters (number of layers, number of neurons of each layer, the introduction of fully connected layers for the LSTM and CNN, and the number of filters and their size for CNN), aiming to find the best model. 

Data processing and model implementation were performed using MATLAB^®^ R2019a (MathWorks Inc., Natick, MA, USA) with a machine with an Intel Core i7-3630QM with a maximum clock rate of 2.4 GHz.

### 2.7. Models’ Evaluation

From the eight participants, we randomly left aside one participant for testing the accuracy of the best model regarding unseen data (test dataset), leaving the remaining seven participants for the training and validation process (training dataset). Given the user-specific variability of the energy expenditure, we implemented a leave-one-subject-out cross-validation technique (LOOCV) with the number of epochs fixed to 1000. This technique enabled us to identify the best model and the respective hyperparameters. 

To assess the performance of our models, the following metrics were used, taking the energy expenditure as the ground truth: (i) MSE, (ii) root-mean-squared error (RMSE), (iii) normalized MSE (NMSE), and (iv) Spearman’s correlation coefficient (SCC). The MSE and RMSE were calculated to give a relative accuracy regarding the neural network’s estimation. A value closer to 0 indicates that the neural network performs well. The SCC and NMSE were also evaluated to assess the correlation between target and estimation. A value closer to 1 indicates a perfect fit between both. Additionally, we computed the Bland–Altman plot (enables to compare two measurement techniques, in which the difference between two techniques is plotted against their average) and the linear regression plot with the respective coefficient of determination (R^2^) to assess, respectively, differences and linearity between energy expenditure estimation using indirect calorimetry or the neural network.

## 3. Results

### 3.1. Deep Learning Regression Models Comparison

Table 2 presents the MSE, RMSE, NMSE, and the SCC for the best architectures of LSTM and CNN, and their respective hyperparameters. Regarding the LSTM, we verified that one cell with 150 neurons yields the best model considering this network, with an average MSE of 0.25 W/kg and RMSE of 0.45 W/kg. The NMSE was high (NMSE = 0.67), which shows that this network performs reasonably well in estimating energy expenditure. Increasing the number of neurons to 200 or the number of LSTM layers did not entail an improvement in the energy expenditure estimation, presenting a higher value of MSE (≥0.28 W/kg, RMSE ≥ 0.51 W/kg). The introduction of a fully connected layer also did not improve the tool’s power estimation, presenting a MSE of 0.32 W/kg and RMSE of 0.54 W/kg. Figure 3 shows the best architecture found for the LSTM in this study.

Regarding CNN, the best model was composed of two convolutional layers of 8 filters of size 10, followed by a fully connected layer of 10 neurons, as shown in Figure 4. The best model presented an average MSE of 0.14 W/kg and RMSE of 0.36 W/kg, being more accurate when compared with the LSTM. With the hyperparameters tuned, it was verified that one convolution layer instead of two, regardless of the filter size (5, 10, or 15) or the number of filters (8, 16), entailed a higher MSE (≥0.19 W/kg, RMSE ≥ 0.41 W/kg). Moreover, we observed that increasing the number of neurons of the fully connected layer to 50 did not improve the estimation’s performance (MSE = 0.22 W/kg, RMSE = 0.44 W/kg).

### 3.2. Best Model Performance

#### 3.2.1. LOOCV Performance Analysis

Figure 5A illustrates the energy expenditure estimation considering the best model found in this work (CNN with hyperparameters of Table 2) for the worst, medium, and the best prediction subject of the LOOCV algorithm. Figure 5A shows that the CNN has a good capacity to estimate the steady-state energy expenditure and to discriminate between basal and walking energy expenditure. This neural network was able to associate higher energy when the participants were walking, as we can inspect in Figure 5A. This result is supported by a high and positive SCC (above 0.80).

These results are in accordance with Figure 5B,C, illustrating, respectively, the Bland–Altman and the linear regression plots for the worst, medium, and the best subject of the LOOCV. From Figure 5B, we verify that most of the error’s dispersion is within the 95% confidence interval, with a bias closer to 0. The worst subject presented a positive bias of 0.48 W/kg, the medium subject presented a negative bias of −6.5 × 10^−2^ W/kg, and the best subject presented a positive bias of 6 × 10^−2^ W/kg. From Figure 5C, the subject with the worst prediction presented an R^2^ of 0.73, with the linear fit (Figure 5C, black line) slightly deviated from the ideal line (Figure 5C, red line). The subject with medium prediction and the subject with the best prediction presented a best linear fit, closer to the ideal line, with R^2^ of 0.76 and 0.95, respectively. The mean value of R^2^ was found to be 0.79 (${\bar{R}}^{2}$ = 0.79 ± 0.12), indicating a high fit regarding the target energy expenditure.

#### 3.2.2. Performance Analysis in New Data

We also evaluated the best model performance regarding unseen data (test dataset). It was verified that it estimated reasonably well the test dataset, as illustrated in Figure 6, presenting an MSE of 0.19 W/kg and an RMSE of 0.44 W/kg. The NMSE and the SCC were positive and above 0.7.

#### 3.2.3. Dependency on Gait Speed and Walking Condition

We investigated if the CNN accuracy depends on gait speed and walking condition (i.e., assisted vs. non-assisted walking). For this analysis, the MSE for each of the three gait speeds was evaluated and considering the trials in which the user walked with and without the ankle-foot exoskeleton. The MSE values were used to create a heatmap, showing the error’s dispersion, as illustrated in Figure 7. By analyzing Figure 7, the CNN presented similar values of MSE for both assisted and non-assisted walking and considering the three gait speeds. The MSE was considered low (≤0.17 W/kg) for all conditions.

## 4. Discussion

This work presents and validates a deep learning-based tool for steady-state energy expenditure estimation without relying on indirect calorimetry. Although gold standard, indirect calorimetry is an expensive technique, besides being cumbersome and slightly invasive for persons with motor disabilities. Furthermore, it is not adequate for human-in-the-loop control strategies since it takes more than three minutes to reach the steady-state condition, which is a long time for real-time optimization problems. Our investigation sought to find alternative methods to estimate energy expenditure through smaller, ergonomic, and clinically validated sensors, namely inertial, EMG, and heart rate sensors.

To estimate the subjects’ energy expenditure, we implemented and compared two deep learning approaches, namely the CNN and LSTM neural networks. To find the best model that entails a lower estimation error, we evaluated some network’s hyperparameters with a LOOCV algorithm. 

### 4.1. Comparative Analysis of Deep Learning Regression Models

Regarding the LSTM, we verified that increasing the number of neurons until 150 promoted an improvement in the network’s performance. However, we verified an increase of more than 13% in RMSE (∆_error_ ≈ 0.06 W/kg) when overcoming 150 neurons or when introducing another LSTM cell. The introduction of a fully connected layer after the LSTM cells did not improve the network’s performance, resulting in an increase of 19% in RMSE (∆_error_ ≈ 0.08 W/kg). The best model yields one LSTM cell with 150 neurons.

Considering the CNN, we verified that one convolutional layer was not enough to attain the best energy expenditure estimation, resulting in RMSE 13% higher than the best architecture found with two convolutional layers (∆_error_ ≈ 0.05 W/kg). Moreover, increasing the number of neurons of the best CNN’s fully connected layer did not improve the power’s estimation (RMSE increased 21%, ∆_error_ ≈ 0.08 W/kg). The best model yield two convolutional layers of 8 filters each and one fully connected layer with 10 neurons. 

By comparing the two deep learning architectures, we verified that the CNN yields a more accurate estimation in all metrics assessed in this work. It presents an improvement of 44% in MSE (∆_error_ ≈ −0.11 W/kg), 20% in RMSE (∆_error_ ≈ −0.090 W/kg), and 18% in NMSE (∆_error_ ≈ 0.12, with values closer to 1.0, indicating a better fitting). Furthermore, the CNN achieved better generalization than the LSTM, since a lower standard deviation was obtained for all metrics. This neural network was reported in ref. [22] as suitable for energy expenditure estimation when compared with an MLP.

### 4.2. Detailed Analysis of the Best Model

This study demonstrates the suitability of CNN to accurately estimate the subjects’ steady-state energy expenditure based on inertial, EMG, and heart rate sensors.

A deep analysis of Figure 5 reveals that the CNN was able to catch relevant information in the inputs to increase its generalization to different subjects and walking conditions, supported with the low standard-deviation observed. By analyzing Figure 5A, our model are revealed to be accurate in distinguishing the basal from the walking energy expenditure, and it was sensitive to changes in gait speed and walking condition, since an SCC above 0.85 was achieved, indicating that both target and estimation share the same monotony. Figure 5B, which illustrates the Bland–Altman plots, also highlights the suitability of the proposed CNN. The errors between target and estimation were within the 95% confidence interval and the bias was small and close to 0. For the subject with worst estimations, a positive and higher bias (0.48 W/kg) was expected since an underestimation is visible in Figure 5A, especially during the walking condition. For the subject with medium estimations, the CNN slightly overestimates the target energy expenditure and, thus, it would be expected a negative bias. However, this overestimation is not valid for all trials, explaining the negative, yet close bias to 0 (−6.5 × 10^−2^ W/kg). Regarding the subject with best estimation, the CNN achieved a perfect fit, explaining the low bias of 6 × 10^−2^ W/kg. These results are supported by Figure 5C, where high linearity is observed (${\bar{R}}^{2}$ = 0.79). Therefore, the steady-state energy expenditure estimation with the CNN is comparable to that obtained with indirect calorimetry.

When estimating energy expenditure for a novel subject, the CNN proved to be reliable. Although the increment of 36% in MSE (∆_error_ ≈ 0.05 W/kg) and 22% in RMSE (∆_error_ ≈ 0.08 W/kg), respectively, when compared to the LOOCV algorithm, these values are within the range observed in Table 2 (0.14 ± 0.1 and 0.36 ± 0.13 W/kg for MSE and RMSE, respectively). The test subject presented a higher variation in the baseline energy expenditure in comparison with other subjects (visible in Figure 6), which introduced an additional challenge in estimating energy expenditure. This may explain the higher error when compared to the LOOCV algorithm. Nevertheless, the SCC was high (0.87) and consistent with the LOOCV algorithm (0.87 ± 0.043), indicating that the inertial, EMG, and heart rate sensors provide enough information to estimate accurately the energy expenditure, regardless of gait speed, walking condition, and basal vs. walking energy expenditure. This was also highlighted in Figure 7, which illustrates the MSE dependency on gait speed and walking conditions. Thus, it seems that there is no dependency regarding gait speed, given the similar values of MSE. Further, the MSE mean values were close to 0 (MSE ≤ 0.17), supporting the conclusion that our tool is accurate in estimating energy expenditure. Yet, we verified a slightly better estimation when the subjects were assisted with the ankle-foot exoskeleton. This may be explained by the fact that gait is more controlled when subjects walk with the ankle-foot exoskeleton. Considering these results, the proposed tool was revealed to be versatile for energy expenditure assessment in multiple rehabilitation scenarios.

### 4.3. Related Work

Previous works have shown a good performance of AI algorithms in estimating human energy expenditure. For instance, [19] presented an approach to estimate energy expenditure using IMUs, EMG, HR, and, among other physiological signals, the SpO_2_. The users performed numerous tasks without gait assistance, with a minimum gait speed of 0.6 m/s and a maximum of 2.7 m/s, which resulted in high energy expenditure variation (~10 W/kg). The authors used linear regression models with the best ones presenting errors bellow 1.5 W/kg. Our work innovates by exploring deep learning regressors along with orthotics-based gait assistance considering slow gait speeds, which entailed a lower energy expenditure variation (~4.1 W/kg) when compared to [19]. This lower variation of energy expenditure, conjugated with both assisted and non-assisted gait conditions, may cause an added challenge in the estimation process of our model.

In refs. [20,21], the authors found high correlations between the target and the estimated oxygen uptake by a random forest algorithm and MLP, respectively. Zhu et al. [22] assessed the effectiveness of CNN, MLP, and an activity-specific linear regression model, using inertial, HR, anthropometric data (age, height, weight), and basal metabolic rate. The authors found that the CNN yields the best estimation, resulting in an improvement of more than 30% when compared to the MLP and linear regression model. However, these state-of-the-art approaches may require several minutes (at least 3) to compute the steady-state energy expenditure. Thus, it would entail a higher optimization time for applications with the human-in-the-loop control. Our work innovates over studies from refs. [20,21,22] by (i) estimating the steady-state energy expenditure, which is useful for rapid estimation and reliable use in human-in-the-loop control strategies; (ii) benchmarking two different deep learning regressors, namely the CNN and LSTM, along assisted and non-assisted gait conditions while considering slow gait speeds commonly observed in persons with gait disabilities. 

A recent study estimated the steady-state energy expenditure considering ankle assisted walking at 1.25 m/s [15]. The authors used GRF and EMG sensors as inputs of a linear regression model and an LSTM. They found the LSTM was suitable to estimate the end-user energy expenditure, with an average RMSE of 0.40 W/kg for ankle assisted walking, yielding a better model when compared with linear regression (RMSE = 0.43 W/kg). The results were comparable to those obtained with our work, especially for the LSTM neural network. However, our results suggest that using CNN to estimate energy expenditure may improve the estimation power, given the lower RMSE (0.36 W/kg). Additionally, our work innovates the study [15] by exclusively relying on wearable sensor data and exploring slow gait conditions.

### 4.4. Limitations and Future Insights

The main limitation of our work is the low number of subjects to train the models. Although our trials are long in temporal terms, involving more than 360,000 samples in total to develop the regression models, our algorithm may be affected by the subject’s variability. Even so, an equal gender distribution was guaranteed, which is important to have a more representative population. From a future perspective, we aim to collect more data with different subjects to augment the estimation’s power. Future insights also address the estimator’s selection algorithms, by selecting among data the minimal conjugation of sensors that promotes an accurate estimation and fulfill usability guidelines. Lastly, we aim to collect data with persons that exhibit gait disabilities to assess the differences regarding energy expenditure. The use of transfer learning will endow our models with the ability to estimate the energy expenditure for non-healthy persons.

## 5. Conclusions

This work presents and validates a deep learning tool for steady-state energy expenditure estimation for both assisted and non-assisted gait conditions considering slow gait speeds typically observed in persons with motor disabilities. Our approach relies on the use of deep learning to promote an accurate estimation of the subjects’ energy expenditure using data collected by reliable, ergonomic, and clinically validated on-body sensors. From the cross-validation results, we verified the suitability of CNN to accurately estimate energy expenditure in basal, assisted, and non-assisted walking, resulting in an improvement of 20% in RMSE when compared to LSTM. Therefore, we propose a versatile tool for estimating the energy expenditure in multiple gait conditions (freely human walking and robotics-based gait rehabilitation) and in optimization problems issues in the human-in-the-loop control strategies.

## Figures and Tables

**Figure 1 sensors-22-07913-f001:**
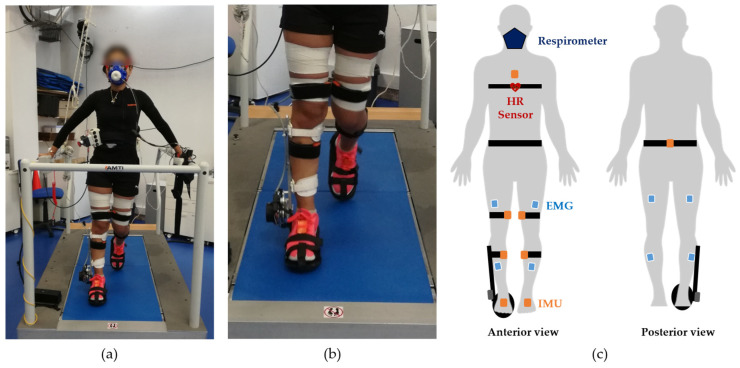
Representation of the setup’s instrumentation on the human body: (**a**) example of a participant during an assisted walking trial, while wearing all sensor setup (kinematic, EMG, and HR sensors and the respirometer); (**b**) zoomed-in view of the ankle-foot orthosis from the H2-Exoskeleton during the walking trial; and (**c**) schematic illustrating the IMUs placement on *sternum*, pelvis, upper leg and lower leg; the EMG sensors on the *vastus lateralis*, *bicep femoris*, *tibialis anterior*, and *gastrocnemius lateralis* muscles; the Polar H10 heart rate sensor at the chest; the respirometer, covering the user’s facial respiratory ways; and the ankle-foot orthosis fixed on the shank and foot segments of the right lower limb.

**Figure 2 sensors-22-07913-f002:**
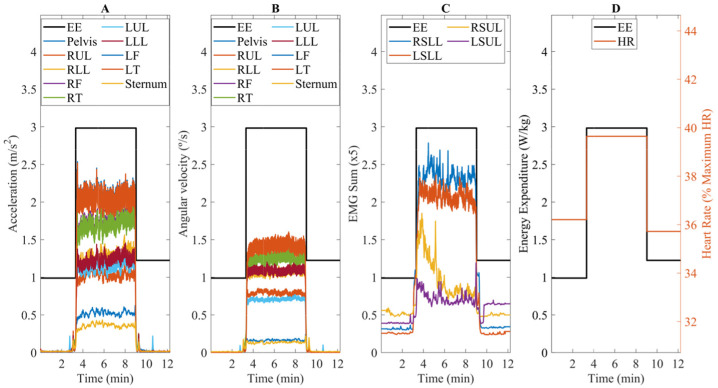
Example of post-processed data used to estimate the steady-state energy expenditure for one random participant: (**A**) acceleration; (**B**) angular velocity; (**C**) EMG sum; and (**D**) HR. EE stands for Energy Expenditure, (R/L)LL for Right/Left Lower Leg, (R/L)UL for Right/Left Upper Leg, (R/L)SLL for Right/Left Sum Lower Leg, and (R/L)SUL for Right/Left Sum Upper Leg.

**Figure 3 sensors-22-07913-f003:**
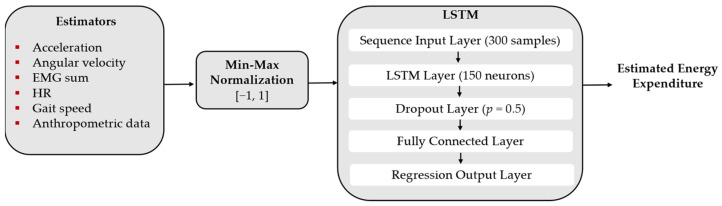
LSTM’s architecture of the final model.

**Figure 4 sensors-22-07913-f004:**
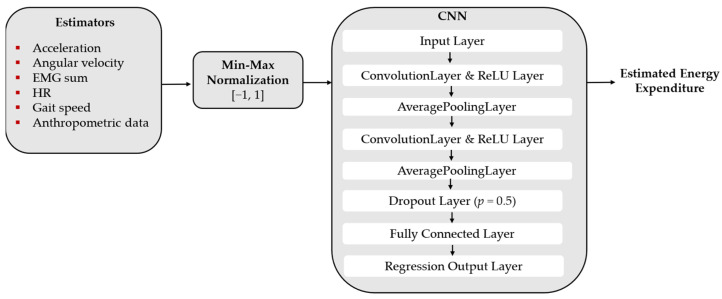
CNN’s architecture of the final model.

**Figure 5 sensors-22-07913-f005:**
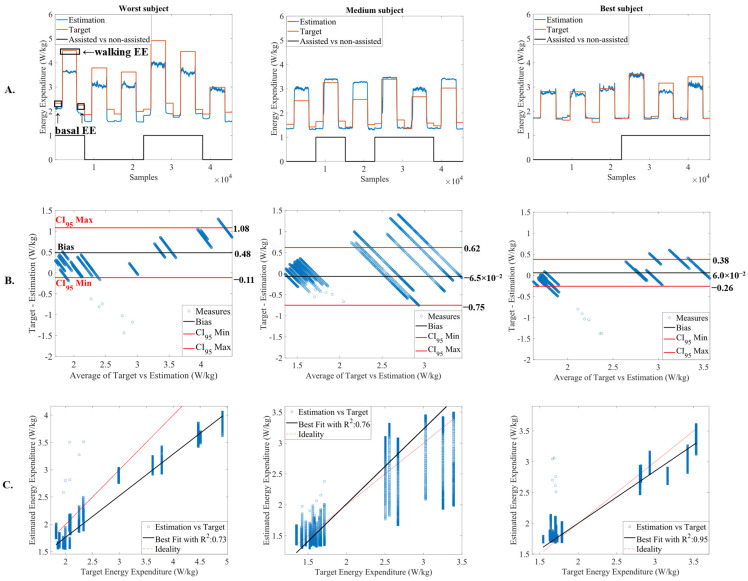
(**A**) energy expenditure (EE) estimation for the worst, medium, and the best prediction subject of LOOCV algorithm, marking assisted and non-assisted gait, and basal and walking energy expenditure; (**B**) Bland-Altman plot of target vs. estimation for the worst, medium, and the best prediction subject; and (**C**) linear regression plot with the coefficient of determination (R^2^) for the worst, medium, and the best prediction subject of LOOCV algorithm.

**Figure 6 sensors-22-07913-f006:**
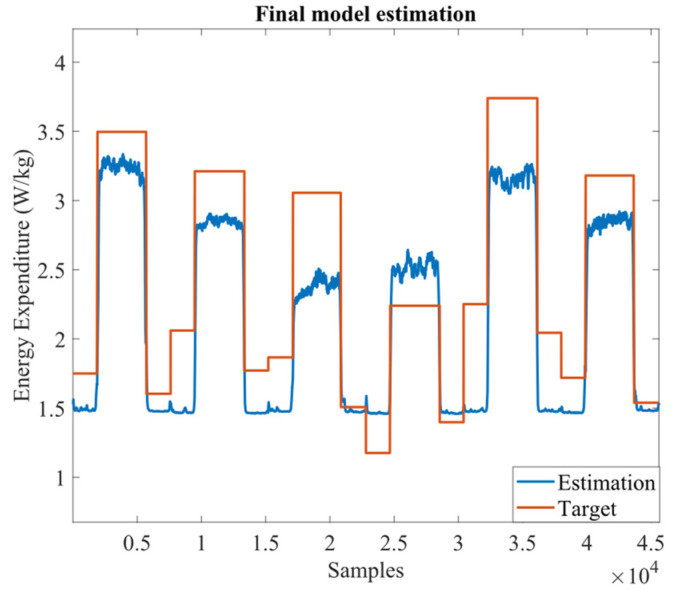
CNN estimation for the novel subject.

**Figure 7 sensors-22-07913-f007:**
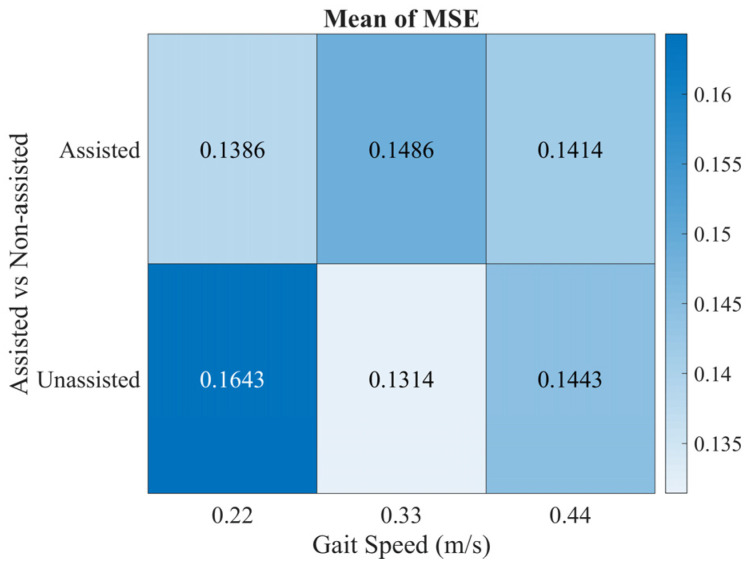
Variation of MSE regarding the gait speed (0.22, 0.33, and 0.44 m/s) and walking condition (assisted vs. non-assisted gait).

**Table 1 sensors-22-07913-t001:** Participants’ anthropometric data and mean (±standard-deviation) regarding gender.

Participant	Gender	Age (years)	Body Height (cm)	Body Mass (kg)
S01	F	27	162	53
S02	M	25	182	77
S03	M	25	181	74
S04	F	26	162	65
S05	F	23	167	61
S06	M	23	170	81
S07	M	23	170	76
S08	F	22	174	59
Male	4	24.0 ± 1.2	176 ± 6.7	77.0 ± 2.9
Female	4	24.5 ± 2.4	166 ± 5.7	59.5 ± 5.0

**Table 2 sensors-22-07913-t002:** CNN and LSTM accuracy in estimating energy expenditure considering the LOOCV algorithm and the test dataset.

Model	Hyperparameters		MSE ^a^		RMSE ^b^		NMSE ^c^		SCC ^d^	
			LOOCV	Test	LOOCV	Test	LOOCV	Test	LOOCV	Test
LSTM	Neurons	150	0.25 (0.22)	N.A.	0.45 (0.22)	N.A.	0.67 (0.23)	N.A.	0.86 (0.035)	N.A.
	Layers	1								
	Batch size	76								
	FC Layer	1								
CNN	Filter size	10	0.14 (0.10)	0.19	0.36 (0.13)	0.44	0.79 (0.12)	0.71	0.87 (0.043)	0.87
	No. filters	2 × 8								
	Batch size	22,800								
	FC Layer	10								

^a^ Mean-Square Error; ^b^ Root-Mean-Square Error; ^c^ Normalized Mean-Square Error; ^d^ Spearman’s Correlation Coefficient.

## Data Availability

Data that support this research will be available upon request.

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
