# Peer review of "Deep Learning-Based Energy Expenditure Estimation in Assisted and Non-Assisted Gait Using Inertial, EMG, and Heart Rate Wearable Sensors"

_sensors, 2022, doi:10.3390/s22207913_

Round 1

Reviewer 1 Report

Dear authors, 

The article entitled “Deep learning-based energy expenditure estimation in assisted and non-assisted gait using inertial, EMG, and heart rate wearable sensors” deals with another way to calculate the energy which used in gait disabilities. It is a very interesting article and can contribute in the field.

The originality in the field is the deep learning approach. The quality of presentation is high and I think is an article which will easily read it from the scientific community. 

The main and crucial problem for me is that, the experimental procedure includes only walking with different speeds and that is very limited. P. Slade et al in their article, uses walking, climbing stairs and biking in their subjects. 

The other main issue, is the small number of participants (it is only 8). That causes a lack of “power” in analysis due to large standard errors. I suggest the authors to include more procedures to experiments and more participants.

Best

Reviewer 2 Report

The overall impression of the paper is mixed for me. Authors need to write clearly and clarify in many places. All comments are in the appendix.

Round 2

Reviewer 1 Report

The authors have responded to my remarks. 

Reviewer 2 Report

The authors have responded to my remarks and I have no other comments.